# Time spent by hospital personnel on drug changes: A time and motion study from an in- and outpatient hospital setting

**Joo Hanne Poulsen**[1]☯*, **Lotte Stig Nørgaard**[1], **Peter Dieckmann**[2,3,4], **Marianne Hald Clemmensen**[5]☯

**1** Social and Clinical Pharmacy, University of Copenhagen, Copenhagen Ø, Denmark, **2** Copenhagen Academy for Medical Education and Simulation (CAMES), Center for Human Resources, Capital Region of Denmark, Herlev Hospital, Herlev, Denmark, **3** Department of Quality and Health Technology, University of Stavanger, Stavanger, Norway, **4** Department of Clinical Medicine, University of Copenhagen, Copenhagen, Denmark, **5** Medicines Information Center, The Hospital Pharmacy, Capital Region of Denmark, Copenhagen NV, Denmark

☯ These authors contributed equally to this work.

* Joo.hanne.poulsen@sund.ku.dk

**Data Availability Statement:** All relevant data are within the manuscript and its Supporting Information files.

## Abstract

### Introduction

Medicines used at Danish public hospitals are purchased through tendering. Together with drug shortage, tendering result in drug changes, known to compromise patient safety, increase medicine errors and to be resource demanding for healthcare personnel. Details on actual resources required in the clinic setting to manage drug changes are unknown. The aim of the study is to explore time spend by hospital personnel in a drug change situation when dispensing medicine to in- and outpatients in a hospital setting in the Capital Region of Denmark.

### Method

A time and motion study, using direct observation combined with time-registration tools, such as eye-tracking, video recording and manual time tracking. Data were obtained from observing nurses and social and health care assistants with dispensing authority while dispensing or extraditing medicine before and after the implementation of drug changes in two clinical setting; a cardiology ward and a rheumatology outpatient clinic.

### Results

Hospital personnel at the cardiology inpatient ward spent 20.5 seconds on dispensing a drug, which was increased up to 28.4 seconds by drug changes. At the rheumatology outpatient clinic, time to extradite medicine increased from 8 minutes and 6 seconds to 15 minutes and 36 seconds by drug changes due to tender. Similarly, drug changes due to drug shortage prolonged the extradition time to 16 minutes and 54 seconds. Statistical analysis reveal that drug changes impose a significant increase in time to dispense a drug for both in- and outpatients.

**Funding:** JHP is doing a co-financed PhD study funded by Amgros I/S, Copenhagen Academy for Medical Education and Simulation (CAMES), The Hospital Pharmacy in the Capital Region of Denmark and the University of Copenhagen. Peter Dieckmann holds a professorship at the University of Stavanger that is funded by an unconditional grant of the Laerdal Foundation (Stavanger, Norway) to the University of Stavanger. The funders had no role in study design, data collection and analysis, decision to publish, or preparation of the manuscript.

**Competing interests:** The PhD project is co-funded by Amgros I/S, Copenhagen, Denmark. However, it is important to emphasize that Amgros had no role in the execution of the study, such as study design, data collection analysis, decision to publish, or preparation of the manuscript. In Denmark the healthcare system is primarily funded by taxes. Public hospitals purchase medicines through tendering and procurement which is a national task carried out by Amgros. Amgros I/S is a noncommercial company owned by the five Danish Regions. On behalf of the Regions, Amgros secures patients access to new and effective treatments and devices, just as a high level of security of supply is ensured and drug relates costs are restrained. The tendering and procurement of medicine takes place in close collaboration with hospital pharmacies, drug committees, the Danish Medicines Council, suppliers and heads of procurement in the Regions. This task would otherwise be maintained independently by the five regions. Therefore, Amgros plays a central role in ensuring supply of medicine and supporting patient safe drug changes in public hospitals, but has no economic interest in the present study. With this competing interest statement, I, Joo Hanne Poulsen, confirm that the co-funding from Amgros I/S to the PhD project, does not alter our adherence to PLOS ONE policies on sharing data and materials.

## Conclusion

Clinical hospital personnel spent significantly longer time on drug change situations in the dispensing of medicine to in- and outpatients in a hospitals. This study emphasizes that implementing drug changes do require extra time, thus, the hospital management should encounter this and ensure that additional time is available for the hospital personnel to ensure a safe drug dispensing process.

## Introduction

Healthcare systems worldwide are under constant economic restraints. With the increasing use of medicine and new expensive medical treatments, healthcare expenses are increasing globally [1]. In Denmark, hospital medicine expenses have increased from DKK 4.3 billion in 2007 to DKK 7.9 billion in 2015 [2]. These expenditures will continue to grow driven by the development of new, highly priced drugs together with the increasing number of patients requiring treatment due to demographic changes [3–6]. Consequently, there is an urgent need to utilize and optimize healthcare resources.

In Denmark, the principles of the healthcare system, primarily funded by tax, are based on a free and equal access to healthcare to all citizens [6]. Medical treatment is free of charge for patients admitted to a hospital, just as some patients get medicine extradited and free of charge for self-administration through outpatient clinics at hospitals [6]. Medicines used at public hospitals are purchased through tendering and procurement maintained by Amgros, a national organization owned by the 5 Danish regions [7]. This entails an annual invitation to drug tenders, and when drugs are procured from a new supplier, a drug change (DC) occurs [3, 7]. This set-up aims to provide the Danish hospitals with medicines at the lowest possible cost [7, 8].

Implementing DCs is a field of growing interest, mainly due to patient safety concerns [9–13] and emerging evidence of large resources being used to implement DCs, especially in relation to drug shortages (DSs) [14–18]. The majority of the literature focusing on DSs are based on surveys (retrospective studies) or on self-reported data [15–18]. A European survey revealed that nearly 50% of all hospital pharmacists reported spending a minimum of 5 hours/week managing DSs [16]. Additionally, self-reported data from Belgium hospital and community pharmacists revealed median times of 109 and 25 minutes respectively spent on managing drug supply problems per week [15, 17]. In the USA, annual labor cost related to managing DSs were estimated as high as US$ 259 million per year in 2019 [18]. Thus, managing DSs and implementing DCs following shortages are continuously causing increased expenses in hospitals. Hence, optimizing resources spend by hospital personnel are of great interest to healthcare systems.

DCs challenge patient safety and an increasing number of medication errors related to DCs have also been reported [9, 12, 13]. Additionally, treatment with a new drug unfamiliar to hospital personnel are challenges described in the literature [11–13]. DCs also generate insecurity among patients who self-administer their medicine at home [9, 19]. Thus, the importance of e.g. nurse-led activities related to patient education, as well as providing information and continuity of care are important indicators for improving adherence to medical treatment in outpatient clinics [20–23].

The focus of this study is the DC situation, meaning the actual changes to new drugs unfamiliar to the dispensing personnel. As already mentioned, DCs can be caused by tendering or

DS, but they can also be initiated by activities related to the implementation of new clinical guidelines [9]. However, regardless of cause or origin, the general term DC will be used throughout this paper.

To the best of our knowledge, no studies exploring time spent on DCs in the clinical setting using direct observations and time-registrations have been published. Additionally, eye-tracking technology is introduced as a time registration tool, where a head-mounted eye-tracker composes Tobii Pro Glasses [24], which allows the capturing of a first person perspective video of the focus of attention [25]. Tobii eye-tracker has demonstrated suitability in research areas such as marketing and consumer research, clinical research and education [24–26]. Thus, this study aims to explore the time spend by hospital personnel in a drug change situation when dispensing medicine to in- and outpatients in a hospital setting in the Capital Region of Denmark.

## Methods

### Setting

The study was undertaken at Bispebjerg and Frederiksberg Hospital (BFH), which is one out of seven main hospitals in the Danish Capital Region with an uptake area of approximately 480.000 citizens. BFH has approximately 80.000 acute admissions and 400.000 outpatient visits each year [27, 28].

Based on an internal list containing the expected DCs due to tender, two clinical settings known to experience several DCs were identified and included in this study by two authors (JHP and MHC) [29]; an inpatient cardiology ward and an outpatient rheumatology clinic.

**Sub-study 1: Cardiology ward (inpatient setting).**  At the 60-bed cardiology ward, nurses and social and health care assistants (SHAs) with dispensing authority manually dispense and administer medicines to hospitalized patients. The majority of the dispensing personnel are nurses, with an average of 15 nurses present during the day shift and 12 nurses during the evening shift [30]. The medicine inventory room at the ward primarily consists of standard medicine assortment with high-frequent medicine adjusted according to its general use, area of specialization and expenditures. Additionally, a smaller amount of non-standard medicine assortment is also available at the ward. In the current study, the focus is on the dispensing of standard medicine assortment as DCs primarily occur within this group. Medicine at the hospital is ordinarily provided at 8 am, noon, 5 pm and 10 pm with dispensing approximately 1 hour prior to administration.

The medicine is dispensed from large original containers into a small beaker. The dispensing process includes 6 steps (Table 1) with step 3–6 of particular interest in the current study, as they are the ones affected by DCs.

**Sub-study 2: Rheumatology outpatient clinic.**  The rheumatology outpatient clinic covers approximately 12.000 patients spread over 32.000 visits each year. The main activities in the

**Table 1.  The dispensing process at the cardiology ward.**

| 1. Look up the patient | Enter the electronic system, locate patient, find medicine list |
| --- | --- |
| 2. Prepare the dispensing | Print patient label, put label on medicine beaker or its lid |
| 3. Searching and fetching | Identify and fetch drug from shelves |
| 4. Barcode scanning | Barcode scan the container as a control measure |
| 5. Dispense the dose | Dispense tablet, capsule, suppository, ready-to-use preparation. |
| 6. Validate and complete | Check the dispensing of prescribed drug |

clinic are diagnoses, treatment and rehabilitation of patients diagnosed with rheumatology [28, 31].

Patients can be either given medical treatment on-site (infusions) or medicine can be extradited for self-administration at home (primarily subcutaneous injections). The latter involves a nurse-led patient consultation consisting of the steps shown in Table 2, all of interest to this study.

## Design

A purposive sampling strategy was applied as the inclusion of clinical settings was based on expected DCs, thus they are assumed to contribute to the purpose of this study [32]. Subsequently, participants from these clinical settings were included using a convenient sampling strategy [32]. The intent of the study was not to collect generalizable data, but to get an insight into two highly relevant medical areas affected by DCs. The design of the study was a time and motion study using direct observations and time-registration tools such as eye-tracking, video recording and manual time tracking to collect data. The participants were followed in real-time during medicine dispensing prior to and after DCs were registered [33, 34].

As the dispensing process at the cardiology ward involves seconds, the use of a refined and detailed data collection technique is required. Thus, the eye tracking was used in this setting. Additionally, the dispensing process at the rheumatology outpatient setting contain other activities and patients, thus, the use of an overall time measurement, such as a stopwatch was more useful in this setting. The use of the details provided by the eye tracking were not relevant in outpatient setting, just as the physical presence of the glasses would induce a considerable disturbance in the patient consultation. Thus, different data collecting technique were used for in- and outpatient settings.

For both clinical settings, a description of the workflow related to the dispensing phase was outlined based on information from regional guidelines [35]. An observational data collecting scheme containing the different time-components for the nurse-led patient consultation was outlined. A pilot study for both sub-studies was carried out beforehand and minor adjustments were made accordingly. The actual implementation of DCs is effective from April 1st 2019. Thus, data were collected in a baseline and a comparison period prior to and after April 1st respectively.

The recruitment of nurses and SHAs with dispensing authority took place in close collaboration with the departmental nurse and responsible nurse for medicine at the two settings. Newsletters about the study were send to all relevant personnel, followed by two staff meetings where written and verbal information were provided. Subsequently, participants volunteered to the study through email registration.

**Sub-study 1: Cardiology ward (inpatient setting).** Time registrations were made by direct observation combined with eye-tracking technology (Tobii Pro Glasses 2) [24]. In case of loss of eye tracking data, a Go-Pro camera (Hero 4) was set up in the medication room as a

**Table 2. Steps in nurse-led patient consultation.**

| Preparation | • Access to and reading of the electronic patient record, including check of latest blood sample result<br>• Look up and reading about medicine (if necessary) |
|---|---|
| Consultation | • Professional aspects of treatment (effect, side effect, questions etc.)<br>• Practical aspects of the treatment (ordering new blood tests, appointment making, a new consultation time etc.) |
| Other | • Dispensing and medicine extradition for 8–12 weeks<br>• Documentation in the electronic patient record |

recording back-up. The use of direct observations of participants gave details to the dispensing process, which was subsequently used to extract relevant data from the eye-tracking recording.

The participants were asked to perform their usual dispensing routine while wearing recording eye-tracking glasses, which enabled a focus on participants' realistic behavior [36]. Eye-tracking combined with direct observation provided insight into the many nuances of the general dispensing phase and the many components in the available IT-system in hospitals. The study focus was the dispensing steps 3–6 (Table 1): searching and fetching; barcode scanning; dispense the dose. Through eye-tracking recordings and observational notes, the different elements in the dispensing process were divided into time intervals of interests using compatible software to Tobii Pro Glasses. More specifically, in the software the duration of the entire dispensing process was displayed in one timeline (from beginning to end of the recording) and from there, time spent on step 3–6 in the dispensing of each drug was divided into individual time intervals (segments). Potential irrelevant time aspects to this study were identified and subsequently discarded, e.g. step 1–2 in the dispensing process, disruptions from colleagues or typing irrelevant information into the electronic system etc. The time division, also referred to as coding in this study, was performed immediately after each observation. An example of the coding of relevant and irrelevant time intervals related to medicine dispensing in the cardiology ward for a particular scenario covering the dispensing process step 3–6 for Hexalactone has been provided in Fig 1.

At the cardiology ward, each nurse and SHA was responsible for dispensing medicines to 4–5 patients at each of the four administration periods from 8 am, noon, 5 pm and 10 pm (day and evening shift). Together with eye-tracking, the use of direct observation with continuous time registration on specific dispensing elements requires a 1:1 participant to observer ratio, thus, only one nurse or SHA could be followed a day and an evening shift respectively.

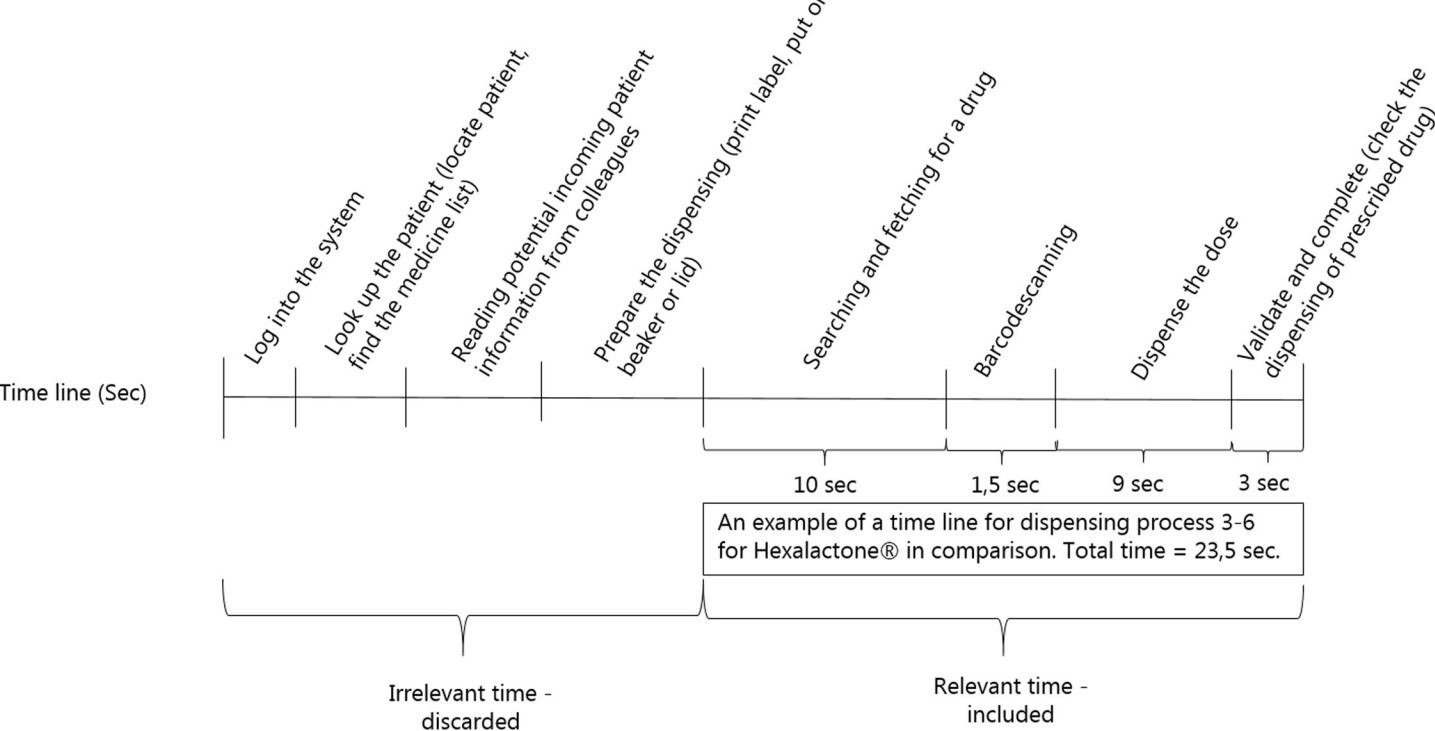

**Fig 1. Coding of relevant and irrelevant time intervals–an example.** An example of coding of relevant and irrelevant time intervals related to medicine dispensing at the cardiology ward for a particular scenario covering the dispensing process step 3–6 for Hexalactone.

Nine nurses and one SHA with dispensing authority were observed in two shifts (day or evening) in a 1) baseline period consisting of five consecutive weekdays in March 2019 and 2) a comparison period with many assumed DCs in April and May 2019. The comparison phase took place over a two month period, as the same 10 participants from baseline in preferably the same shifts (day or evening) were observed. JHP registered all observations.

**Sub-study 2: Rheumatology outpatient clinic.** The time registration was performed by the use of a stopwatch on a mobile phone, while joining nurse-led patient consultations as an observer. The different phases in the consultation was marked using the "round-bottom" in the stopwatch function.

*Metex pre-filled syringe (PFS).* The observational focus was Methotrexate subcutaneous injection administered in pen and pre-filled syringe (PFS). From the tendering it was known that Methotrexate marketed as Metex® PFS (all strengths) would change to Injexate® PFS (all strengths). Of the 12.000 patients from the outpatient clinic, this specific generic change was estimated to affect around 1250 to 1500 patients [31]. The treatment with 15, 20 and 25 mg was specifically studied, as these strengths are the commonly most used (Fig 2).

Eight nurses employed at the outpatient clinic were recruited for direct observations for five consecutive weekdays from 8 am to 2.30 pm (4.30 pm on Thursdays) in a baseline period in March 2019. Both Metex pen and PFSs were included in the baseline registrations, since both represent the patients' accustomed treatment (both when it comes to name, looks and device). DCs were observed in a comparison period for five consecutive weekdays in April 2019 and only included Injexate PFS. First author (JHP) performed the observations together with six pharmacy internship students, who had received guidance on how to observe and complete the observational data collecting scheme.

A follow-up phase was scheduled (approximately) 3–6 months after both Injexate® PFS 15, 20 and 25 mg were well-implemented. The purpose was to explore time spent on nurse-led patient consultation and extradition of Injexate PFS after patients had become more familiar with the treatment, hence, estimating the expenses related to the change from Metex to Injexate PFS. However, Injexate PFS (all strengths) went into a shortage period (July–late November 2019), thus, the follow-up took place between February and March 2020.

*Metex pen.* Metex injections were available in pen and PFS. Whether the patients were treated with one or the other depended on patient and/or doctor preferences. Metex pen (all strengths) was not supposed to undergo a DC due to tender, thus, no changes for patients in treatment with Metex pens were planned. However, Metex pens 25 mg went into DS while data were collected in March and April 2019. Thus, time spent on nurse-led patient consultation concerning DS of Metex pens 25 mg was also included in the study (Fig 1).

**Economic impact.** To estimate the economic impact on time spent on DCs at the cardiology ward and outpatient setting, cost data used in this study were extracted from the Danish Medicines Councils guideline "Valuation of unit costs" [37]. The guideline contained specific

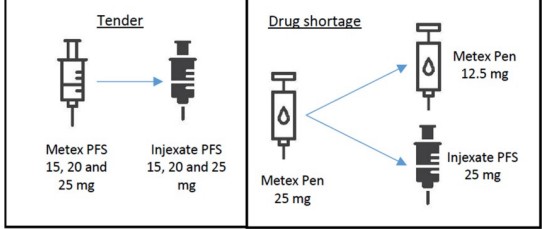

**Fig 2. Drug changes due to tender and drug shortage.** Drug changes due to tender (Metex PFS change to Injexate PFS) and drug shortage of Metex Pen 25 mg with alternatives (Injexate PFS 25 mg or 2x Metex pen 12,5 mg).

unit costs related to e.g. personnel expenses and references for calculating unit costs. The data will be converted to USD using the 2019 average exchange rate (USD 1 = DKK 6,67) [38]. All cost values in cents will be rounded off to the next integer.

At the cardiology ward, the economic impact was estimated at ward level based on: 1) the average number of dispensed drugs; 2) the percentage of drugs undergoing DCs and 'semi-drug changes'; 3) the average number of dispensing nurses in day and evening shift; 4) the time spent on DCs; and 5) conversion into total expenses in USD.

At the rheumatology outpatient clinic, the economic impact was calculated per outpatient clinic based on: 1) the number of patients affected by the DC, 2) the time spent on the DC and 3) conversion into total expenses in USD.

## Ethic

The study is a non-interventional, descriptive study using direct observation, time-registration and eye-tracking technology. The ethic committee in the Regional Committee was consulted, and it waived a formal review of the study (Journal no: H-18053511). Participants were informed both verbally and in writing about the nature of the study, the publication plans and that they could withdraw from the study at any time without further consequences. All participants signed a consent form and were anonymized in the study.

This study did not assess the incidence of medication error, but only focused on time spent on medicine form dispensing or extradition. However, if any dispensing error occurred during the observations, the observer would note and react to this.

## Data analysis

Time spent on DCs at the cardiology ward was extracted using "iMotion software" compatible to Tobii Pro Glasses 2. In situations where recording back-up from the Go-pro camera (Hero 4) were relevant, VLC media player 3.0.6 (January 9, 2019) was used to extract data. Time registrations from both sub-studies were registered in Excel 2016. All statistical analyses were performed using IMB SPSS Statistics 25.

Normal distributions of data were tested using the Kolmogorov-Smirnov Test for sub-study 1 ($> 50$ samples) and the Shapiro-Wilk Test for sub-study 2 ($< 50$ samples). Data were subsequently analyzed using a one-way ANOVA followed by Tukey HSD post hoc test. A Kruskal–Wallis test was used for analysis where normal distribution could not be assumed. Statistical significance was accepted at $P<0.05$.

## Results

Below, data from the cardiology ward are displayed in seconds as one drug dispensing takes less than a minute. Data from the rheumatology outpatient clinic are displayed in minutes as a nurse-led patient consultation takes several minutes.

### Sub-study 1: Cardiology ward (inpatient setting)

Ten nurses were included from the cardiology ward of which nine were women. Working experience varied from less than one year to more than 5 years.

In the baseline period, 199 drugs were dispensed of which 161 were included as they belonged to the standard medicine assortment. Excluded recordings (N = 38) covered dispensing of non-standard medicine assortment, measurement or preparation of dose as they include additional dispensing elements and thus, is not of interest to the study.

In the comparison period, 227 drugs were dispensed of which 35 were included, as they underwent a 'semi-drug change' (N = 26) or an implemented drug change (N = 9). Excluded recordings (N = 192) covered drug dispensing where no changes were registered from baseline to comparison, including standard and non-standard medicine assortment and measurement or preparation of dose. The term 'semi-drug change' is not standardized, but used in this study to cover situations where there is a discrepancy between the prescribed drug (e.g. Bisacodyl marketed as Toilax®) and the physical available drug (e.g. Bisacodyl marketed as Dulcolax®) in the medicine inventory room. This phenomenon emerged when the hospital pharmacy supplies the drug procured from a new supplier and the electronic system is changed accordingly, but some of the "old" drug is still available in the inventory room. Thus, the "new" prescribed drug is not immediately available for dispensing, but is already up for prescription. An implemented DC refers to the situation where the "new" prescribed drug is the drug available for dispensing.

Table 3 displays the mean time spent in seconds per drug dispensed in the cardiology ward. As shown in Table 3 a 'semi-drug change' prolonged a drug dispensing with approximately 8 seconds whereas an implemented DC took approximately 5 seconds longer than a drug dispensing under baseline conditions.

## Sub-study 2: Rheumatology outpatient clinic

Eight nurses were included from the rheumatology outpatient clinic of which seven were women. Working experience varied between 1 year and up to more than 10 years.

On average, a nurse-led patient consultation including drug dispensing took 8 minutes and 6 seconds under baseline conditions. In comparison, the same process took on average 15 minutes and 36 seconds and 16 minutes and 54 seconds for DCs due to tendering and DS respectively. Thus, a nurse-led patient consultation took significantly more time when it included a DC. A follow-up after the implementation of Injexate revealed that time spent on nurse-patient consultation returned to baseline levels (Table 4).

**Economic impact.** From the Danish Medicines Councils guideline "Valuation of unit costs", the unit cost for a nurse in hospital is 84 USD/hour, including overhead expenses in hospital and ward, time for non-patient related tasks, breaks and absences other than holiday (e.g. competency development) [37].

*Cardiology ward (inpatient setting).* Data were obtained from the observations at the cardiology ward and based on the number of dispensed drugs in baseline and comparison, the average number of dispensed drugs is 10 drugs per nurse per shift. The percentage of drugs undergoing 'semi-drug changes' and DCs were 11.5% and 4% respectively. Thus, the time spent per nurse in one shift on 'semi-drug changes' = 9 seconds (10 drugs x 11.5% x 7.9 seconds) and DCs = 2 seconds (10 drugs x 4% x 5.4 seconds), a total of 11 seconds per nurse per shift.

At the cardiology ward, 15 nurses and SHA's dispenses during the day and 12 nurses and SHA's during the evening, leaving the extra time spent on 'semi-drug changes' and DCs at the

**Table 3. Mean time (seconds) spent per drug dispensed (N).**

| Cardiology ward | Type | N (no) | Mean (sec) | Standard Deviation (sec) | Median (sec) | Minimum (sec) | Maximum (sec) |
|---|---|---|---|---|---|---|---|
| Baseline | Standard assortment | 161 | 20,5 | 10,8 | 17,4 | 8,6 | 72,2 |
| Comparison | Semi- drug changes | 26 | 28,4* | 16,3 | 22,5 | 14 | 90 |
| | Drug changes (implemented) | 9 | 25,9 | 15,6 | 18,2 | 11,2 | 59,2 |

*P = 0.003 (Kruskall-Wallis test). Pairwise comparison showed a significant difference between baseline and semi-drug changes (P = 0.001).

**Table 4. Mean time spent in minutes per nurse-led patient consultation (N).**

| Outpatient Clinic | Type | N (no) | Mean (min) | Standard Deviation (min) | Median (min) | Minimum (min) | Maximum (min) |
|---|---|---|---|---|---|---|---|
| Baseline | Metex pen or syringe | 14 | 8 min 6 sec | 2 min 36 sec | 8 min 6 sec | 4 min 24 sec | 12 min 48 sec |
| Comparison | Injexate syringe | 9 | 15 min 36 sec* | 5 min 42 sec | 16 min 24 sec | 7 min 6 sec | 27 min 54 sec |
| Follow-up | Injexate syringe | 5 | 8 min 54 sec | 3 min 24 sec | 9 min 24 sec | 4 min 18 sec | 13 min 36 sec |
| Drug shortage | Injexate syringe 25 mg<br>2x Metex pen 12,5 mg | 7 | 16 min 54 sec* | 4 min 48 sec | 19 min | 8 min 42 sec | 21 min 6 sec |

*P< 0.001 (one-way ANOVA followed by Turkey HSD post hoc test). Pairwise comparison showed a significant difference between baseline and 1) comparison (P = 0.001), 2) drug shortage (P<0.001).

cardiology ward to 303 seconds per day (27 nurses x 11 seconds). This corresponds to expenses of 7 USD per day at the cardiology ward. Assuming that 'semi-drug changes' and DC implementation affect the dispensing time for a period of 4 weeks, the cardiology ward ends up spending a total of 196 USD.

*Rheumatology outpatient clinic*. Implementing one DC per patient consultation will take 7 min 30 seconds longer (comparison minus baseline), corresponding to 10.5 USD. Thus, with 1250–1500 patients in Methotrexate subcutaneous injection, the extra time spent on DCs for these patients adds up to a minimum of 156 hours and 15 minutes and a maximum of 187 hours and 30 minutes, corresponding to expenses for one rheumatology outpatient clinic of 13.125–15.750 USD.

## Discussion

When experiencing implemented DCs or 'semi-drug changes' in the inpatient setting, the extra time spent on dispensing one drug by hospital personnel was on average 5 and 8 seconds respectively. The statistical analysis reveals a statistically significant difference between baseline and 'semi-drug changes'. In the outpatient clinic, DC due to tender and shortage prolonged the patient encounter significantly by 7 minutes and 30 seconds and 8 minutes and 48 second respectively. Each sub-study will be discussed after a general discussion of DCs.

In this study, the observed DCs are considered to be of a more simple character (generics) compared to more complex changes related to situations involving therapeutic substitution, unlicensed medicine and differences in drug strength [9–12]. Other studies describe general aspects of DCs, including potential patient safety consequences, but they do not involve time registrations related to the changes [9–12]. From the literature, time studies concerning DCs due to shortage exist in smaller numbers, but they are based on self-reports or surveys of time spent on managing DSs on a weekly basis [14–18]. Self-reported data from Belgian hospital pharmacists revealed median times of 109 minutes per week spent on managing drug supply problems per week [17]. Additionally, surveys from European hospital pharmacists report spending a minimum of 5 hours/week, whereas it is estimated that U.S. pharmacy staff spent as much as 8.6 million personnel hour of labor per year on DSs across all U.S. hospitals [16, 18]. However, these results are entirely based on reports from hospital pharmacists and hospital managers, with no inclusion of the clinical setting and the hospital personnel. Our study, on the other hand explore time spent on DCs by hospital personnel in two clinical settings, which makes it is difficult to compare our findings with the existing hospital pharmacy literature.

### Sub-study 1: Cardiology ward (inpatient setting)

An interesting finding was the discovery of the 'semi-drug change' characterized by a discrepancy in prescribed drug and available drug in the inventory room. Significant difference was

found for 'semi-drug changes' compared to baseline and although no significant difference was found for implemented drug changes, extra time (5 seconds) was registered in the comparison period. The extra time used on a drug dispensing could be due to hospital personnel spending time on searching for a particular prescribed drug. This is a subtask that reportable could end up being time consuming for the hospital personnel [25], just as changes in trade names, labels and/or physical packaging unfamiliar to the hospital personnel are other potential causes reported in the literature [9–11].

Furthermore, a discrepancy in prescribed drug and available drug in the inventory room might lead to medication errors. This has been reported in a Norwegian study where 42 out of 100 nurses experienced medication errors related to generic substitution, because doctors had failed to prescribe from revised drug lists owing to DCs caused by tender [13]. Additionally, 'semi-drug changes' could lead to omitted and/or delayed treatment, as nurses may be unaware of synonymous trade names (generics) between the prescribed drug and the available generic drug in the medicine room [9,11,12]. One way of overcoming some of the consequences would be to use generic (pharmacological) names in the entire medication process instead of trade names. Thus, there will be a congruency between names used in prescribing and dispensing in hospitals [13, 39]. However, drugs with the same generic names, but different formulations could lead to look-alike/sound-alike errors [9, 12, 39]. Other reported ways of overcoming challenges related to 'semi-drug changes' are to involve healthcare and regulatory authorities to ensure safer drug names, registration of the same dose units and avoid look-alike packaging [9, 12, 13]. More technical solutions, such as automated medicine cabinets, barcode scanning and updating electronic prescribing systems have also been mentioned in the literature [9, 12, 13]. However, this study only focused on time spent on drug dispensing and did not involve the incidence of omitted and/or delayed treatment.

No follow-up at the cardiology ward took place and thus, claiming that the extra time spent on DCs are related to immediate expenses due to tender are not possible. However, the follow-up data from the rheumatology outpatient clinic indicates that the extra time observed in relation to DC diminished over time and returns to baseline. Thus, one could assume a similar trend at the cardiology ward as the nurses and SHA become more familiar with dispensing of the implemented drugs.

## Sub-study 2: Rheumatology outpatient clinic

The nurses at the rheumatology outpatient clinic spent almost twice as much time managing the change from Metex PFS (baseline) to Injexate PFS (comparison), but at follow-up, the extra time had diminished and the consultation time resembled baseline data. Thus, the additional time spend on changing from Metex to Injexate highlights the immediate expenses related to a specific DC due to tender. Additionally, twice as much time managing the DC due to DS of Metex pen was found.

Possible explanations for the significant time differences are various. When drugs change, nurses are insufficiently prepared for the changes. Primarily, they become aware of DCs when the actual change is implemented with no time to inform or prepare patients beforehand. Even though the route of administration and dosage form are similar for the alternative drug, a change in trade name and/or packaging are potential components compromising patient safety [9, 10, 40]. Additionally, the physical appearance of the device itself (syringe injection) may look or even feel different to inject/administer, thus, patients might feel insecure using it [9]. Such situations require a high degree of information, communication and patient involvement if mistakes or non-compliance are to be avoided [20, 21, 40]. From the literature, nurse-led care is found to provide effective patient education and information, as well as access and

continuity of care. This is directly associated with higher satisfaction amongst patients, leading to an increase in adherence to medical treatments [20, 22, 23]. Additionally, despite obvious time pressure during nurse's working hours, one study reported that it did not hinder the nurses from providing satisfactory emotional support and answering questions from patients, which was perceived as a genuine interest in the well-being of the patient [23]. Thus, the significant time difference for nurse-led patient consultation related to DCs can be considered necessary and valued to patients.

## Economic impact

Scrutinizing DC data at the cardiology ward reveal that drug dispensing only seems to be slightly more time consuming (5.4 to 7.9 seconds). However, recalculating and upscaling the results to cover the entire cardiology ward end up in a scale considered relevant in relation to the time used and expenses spent on the drug dispensing process. At the cardiology ward, a total of 27 nurses and SHA's cover the dispensing process in both day and evening shifts. Estimating that 'semi-drug change' and DC affect dispensing time in a period of 4 weeks leads to a calculated extra time of 2 hours and 22 minutes or 196 USD spent on the assumed 'semi-drug change' and DC at this particular cardiology ward. This extra time and expenses does not immediately seem to have a clinical and economic impact, especially not on a daily basis with extra time corresponding to 303 seconds and 7 USD. However, the size of the cardiology ward is assumed to cover a standard medical ward in the Capital Region of Denmark. Thus, multiplication of the expenses with the total number of medical wards will show a considerably amount of additional expenses spent locally, regionally or even nationally. It should be emphasized that the estimated 4 weeks period with 'semi-drug change' and DC are based on assumptions, since the actual duration of the periods are unknown.

Looking at the extra time spent on one DC from Metex to Injexate in one rheumatology outpatient clinic, the impact of additional expenses are considerable, 13.125–15.750 USD. This corresponds to extra time in the range of 156 hours and 15 minutes and 187 hours and 30 minutes, affecting 1250–1500 patients. Thus, multiplication of the expenses with the number of rheumatology outpatient clinics will show a considerably amount of additional expenses spent used for this kind of drug change. Further, another relevant point of discussion is the estimated consequences of DCs in the clinical setting. Assuming that 12 nurse-led patient consultations are generally scheduled on a daily basis, based on our findings, implementing one DC per patient consultation will take 7 min 30 seconds longer (comparison minus baseline), which sums up to 90 minutes extra time spent on consultations per day. As a result, less nurse-led patient consultations during the implementation phase can be scheduled, which is a noticeable disadvantage for the clinical practice. Thus, implementing DCs may indeed affect the workload of nurses, but it is difficult to estimate the exact consequences for each outpatient clinic as it will depend on the contextual and organizational setting.

A more general discussion relates to whether or not the DC expenses compensate for the gained savings associated with tender and procurement. The incoming bids for tender and actual procurement savings are unknown to the research group, thus, a direct comparison cannot be made. Further studies should elucidate the economic consequences, including all relevant expenses related to the tendering and procurement process and the implementation of DCs. The details of our study show how much extra time a DC in two clinical settings requires and that this extra time spent in an outpatient setting decreases over time. However, a potential explanation for the decrease of time needs further research beyond this study, but in a restrained healthcare system. Extra time is not only money, but also additional pressure on the hospital personnel with potential risks of errors.

## Strengths and limitations

One strength of this study is the use of direct observation in real-time combined with collecting quantitative data, using eye-tracking technology. This enabled us to make a clear distinction and division of the many nuances and aspects related to the IT system in the dispensing process. The precision of the time registrations would have been impossible without eye-tracking data combined with the detailed observational notes containing practical information of what was being prescribed and dispensed, as well as potential time-wasting activities.

Both surveys and self-reports are low cost methods to quantify information about time spend on managing DSs. The major disadvantages with these methods are biases, such as intentional/unintentional over-estimation about time spent in order to exhibit socially desirable behavior or they simply face problems remembering [33, 41]. These biases are met in this study by the use of direct observations.

A limitation with time and motion studies are their resource demanding nature, since a time and motion study and eye-tracking require a 1:1 participant to observer ratio, thus, restricting the number of participants that can be included [33, 41]. Another essential limitation with direct observations and the presence of an external observer is related to the participants' change of behavior while being observed, also referred to as the Hawthorne effect [41, 42]. The extent of the Hawthorne effect is difficult to estimate in the study, but the observer is assumed to have affected the observed participants to a minor extent. Additionally, the physical appearance of the go-pro camera and the eye-tracking glasses mounted to the participant's head have indeed also affected the participants while being observed. The Hawthorne effect in this matter is impossible to avoid, unless the nurses and SHA became accustomed to working with the glasses. Further, as the same participants and observer is present in baseline and comparison, the Hawthorne effect is assumed equally widespread in both data collection periods, although the effect has not been measured.

At the cardiology ward, the same person (JHP) performed all the observations and coding of data. This may have resulted in personal and intra-rating bias, which could potentially affect the reliability of the study [43]. Obviously, having two or more researcher performing the observations and coding independently, discuss their finding or calculating a coder reliability, would enhance the overall reliability and reduce potential personal bias in the study. One way to minimize the potential intra-rating bias is to reduce the risk of "coder-drift", which refers to applying the coding differently due to boredom or fatigue when coding larger data set at once [43]. However, in this study the coding was performed immediately after each observation and thus potentially reducing intra-rating bias [43]. Further, Fig 1 displays the coding frame for the observations hereby increasing the transparency of the coding of included and discarded time. Conversely, having the same person performing the data collection results in less observation bias and reliability issues compared to the outpatient clinic, where pharmacy internship students were involved too. Despite the guidance of students, there might be differences in the perception of the time registrations in the consultation elements. However, these differences are considered to have minor influence on the registered time differences and overall findings.

Consultation with the hospital pharmacy led to the expectation of implementation of DCs at the cardiology ward within 1–3 weeks after April 1st, which was therefore chosen as the comparison period. Even though the actual implementation of DCs were effective from April 1st, relatively few DCs were observed in the cardiology ward in the comparison period (April and May), which potentially affects the reliability of the results. The physical changes in the medicine inventory rooms depend on current surplus medicine stock from hospital pharmacy and the hospital ward/clinic. Thus, it is difficult to pick the right moment for observations in order

to capture the actual implementation of DCs. However, an expansion of the comparison period would most likely have produced more data.

A potential limitation related to the rheumatology outpatient clinic is the differences between patients and their individual knowledge and need for information related to the DC. Thus, some of the patients in either baseline, comparison or shortage situation might not fully reflect the patient population at the rheumatology outpatient clinic. Rathe et al. (2013) found a positive association among patients between a generic change and previous positive DC experiences within the same drug type [40]. In our case, some patients having been in long time treatment with Methotrexate subcutaneous injection may be more acceptable to DCs as they are likely to have experienced DCs earlier. Further, this study focusses on a generic DC due to tender, but one could image that dealing with DC from originator to biosimilar would require extra resources for nursing education and patient information, due to another complexity of the DC. However, another study would be required in order to explore this perspective in details.

Another perspective for the time differences might be due to the diversity among nurses. One might argue that participants who voluntarily signup for the study may be more skilled compared to those who did not sign up. Although nurse characteristics might affect the extra time spent on DCs in the dispensing and patient consultation, it cannot solemnly explain the significant findings in this study, though. By choosing purposive sampling combined with the few data points, one can never claim generalizable results and the external validity may be considered relatively low, but data still provides relevant insights into the resources related to DCs.

## Conclusion

This study revealed that hospital personnel spent significantly longer time on dispensing of medicine when experiencing a 'semi-drug change' in the inpatient setting. Also in the hospital outpatient setting a significant increase in time to extradite medicine was seen after DCs. Until now, the knowledge about resource requirement for managing drug changes has been related to hospital pharmacy personnel, but the present study adds detailed information on how DCs also affect clinical personnel. Furthermore, DCs are also known to challenge the patient safety and it is thus important that the clinical personnel spent extra time and focus on the drug dispensing process to ensure patient safety. The current study does not reveal if the extra time spent on drug dispensing was specifically used on patient safety activities. Future studies are needed to elucidate this perspective. However, this study emphasizes that implementing DCs do require extra time and financial expenses, thus, the hospital management should encounter this and ensure that additional time and resources are available for the clinical personnel to ensure a safe drug dispensing process.

## Supporting information

**S1 Dataset. Time spent by hospital personnel at the in- and outpatient hospital setting.**
(XLSX)

## Acknowledgments

The authors would like to thank all participants, the departmental nurses and the nurses responsible for medicine at the cardiology ward and the rheumatology outpatient clinic for their time, help and contribution to this study. Additionally, a huge thank you to the pharmacy internship students, who helped with the data collection. A special thank goes to the hospital pharmacy at BFH and the hospital pharmacy logistic department in the Hospital Pharmacy,

Capital Region, for contributing with their valuable knowledge, experience in the field and for helping with enabling this study.

## Author Contributions

**Conceptualization:** Joo Hanne Poulsen, Lotte Stig Nørgaard, Peter Dieckmann, Marianne Hald Clemmensen.

**Data curation:** Joo Hanne Poulsen.

**Formal analysis:** Joo Hanne Poulsen, Marianne Hald Clemmensen.

**Funding acquisition:** Lotte Stig Nørgaard, Peter Dieckmann, Marianne Hald Clemmensen.

**Investigation:** Joo Hanne Poulsen.

**Methodology:** Joo Hanne Poulsen, Peter Dieckmann, Marianne Hald Clemmensen.

**Project administration:** Joo Hanne Poulsen.

**Resources:** Joo Hanne Poulsen, Lotte Stig Nørgaard, Peter Dieckmann, Marianne Hald Clemmensen.

**Software:** Peter Dieckmann.

**Supervision:** Joo Hanne Poulsen, Lotte Stig Nørgaard, Peter Dieckmann, Marianne Hald Clemmensen.

**Validation:** Joo Hanne Poulsen.

**Visualization:** Joo Hanne Poulsen, Lotte Stig Nørgaard, Peter Dieckmann, Marianne Hald Clemmensen.

**Writing – original draft:** Joo Hanne Poulsen.

**Writing – review & editing:** Joo Hanne Poulsen, Lotte Stig Nørgaard, Peter Dieckmann, Marianne Hald Clemmensen.

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
