## [Decision Letter · Decision Letter 0]

16 Oct 2020

PONE-D-20-21353

Time spent by hospital personnel on drug changes: A time and motion study from an in-and outpatient hospital setting

PLOS ONE

Dear Dr. Poulsen,

Thank you for submitting your manuscript to PLOS ONE. After careful consideration, we feel that it has merit but does not fully meet PLOS ONE’s publication criteria as it currently stands. Therefore, we invite you to submit a revised version of the manuscript that addresses the points raised during the review process.

We look forward to receiving your revised manuscript.

Kind regards,

Susan Horton

Academic Editor

PLOS ONE

Journal Requirements:

"JHP is a part of a co-financed PhD study funded by Amgros I/S, Copenhagen Academy for Medical Education and Simulation (CAMES), The Hospital Pharmacy in the Capital Region of Denmark and the University of Copenhagen. Peter Dieckmann holds a professorship at the University of Stavanger that is funded by an unconditional grant of the Laerdal Foundation (Stavanger, Norway) to the University of Stavanger. The funders had no role in study design, data collection and analysis, decision to publish, or preparation of the manuscript."

We note that you received funding from a commercial source: Amgros I/S.

Reviewers' comments:

Reviewer's Responses to Questions

**Comments to the Author**

1. Is the manuscript technically sound, and do the data support the conclusions?

Reviewer #1: Partly

Reviewer #2: Partly

2. Has the statistical analysis been performed appropriately and rigorously? 

Reviewer #1: No

Reviewer #2: Yes

3. Have the authors made all data underlying the findings in their manuscript fully available?

Reviewer #1: Yes

Reviewer #2: Yes

4. Is the manuscript presented in an intelligible fashion and written in standard English?

Reviewer #1: Yes

Reviewer #2: Yes

5. Review Comments to the Author

Reviewer #1: The authors have performed an empirical time and motion study to identify the time spent by hospital personnel on drug changes. Data has been collected for inpatient (cardiology wing) and outpatient (rheumatology wing) of a Danish hospital. Have used certain statistical methods to compare the results. However, there are some major issues that have not been addressed in this paper.

1) The data has been collected by a single person in this study (JHP). This definitely leads to personal bias in all the data collection aspects. It would have been better if at least two members were involved in data collection stage.

2) The authors have mentioned that it would not be possible to measure the Hawthorne effect. But this is an experimental study which has been carried out using sophisticated techniques like Tobii eye tracker, Go pro camera etc. Unless the nurses and SHA are trained on these methods, the impact created by these equipment on the behaviour cannot be underestimated.

3) In the cardiology wing, data has been collected in March 2019 (baseline) and April-May 2019 (immediately after DCs are introduced). This is definitely not a good comparison period because the authors have taken a baseline period wherein the nurses and SHAs were very well versed with the existing drugs (March 2019), and the study period of April-May 2019 is a learning phase even for an experienced nurse/SHA. If the data was collected a few months after the drug change (DC), the actual impact of the time could have been assessed and concluded.

4) In pages 7 & 8 of the manuscript, the authors mention how the software used for collecting data of Tobii glass tracker works. "Potential irrelevant times have been discarded". The authors need to explain more on how they decided irrelevant time. Moreover observing times from video and using it for data collection purpose can include 'bias'. Need to be addressed appropriately.

5) The authors should explain why they have used different data collection techniques for inpatient (eye tracker, video etc.) and outpatient (mobile stop watch) setting.

6) I did a quick data analysis for cardiology ward (baseline) and cardiology ward (comparison).The data points for comparison period is very less. More over, the authors have taken an "average of averages" for the time comparison. This is definitely not a correct practise. For instance, even in base line phase average time taken by nurse 5 is 30.7 sec and that of nurse 7 is 14 seconds. So nurses cannot be treated as "homogeneous units" in data analysis. Need more explanation on this.

Reviewer #2: Comments have been pasted from a Word document (attached) as info was lost due to the way the system saves info. Data provided support the conclusions, however the data set is quite limited. The authors have taken a non-biased approach to measuring time impact, using eye-tracking technology rather than just direct observation – this is commendable. The appropriately cite the difficulty with other reports of time impact of drug changes using survey methods, which are affected by recall bias. Unlike most other publications that focus on pharmacy activities, this report focuses on nursing activity and the impact of drug changes. This situation may be more relevant to hospitals that put more drug control in the hands of pharmacy personnel and that use more automated systems, such as Automated Dispensing Units, barcode technology and even unit dose medications. Most North American institutions have medication orders written by generic name and drug selection is not required by nursing personnel. Thus drug changes (in brand provided) do not impact nursing time. The changes reported by the authors were not significant in all categories, with only semi-drug changes causing a significant increase in time spent by nursing – these changes were relatively short-lived, with time spent returning almost to baseline. IN the outpatient setting of rheumatology clinic, time was impacted more significantly, but for a brief interval only for drug changes and also for drug shortage situations, which may occur at any time of year, regardless of contract changes.

The authors suggest (lines 282-286) that system changes such as prescribing using generic names could prevent confusion during drug changes. Inventory management was briefly alluded to as the medication room continued to have stock of the previous brand, leading to incongruence between prescribed drug name and available drug name.

The authors suggest that administrators should be made aware of the need for additional time (resource) to adapt to drug changes, but they have not provided a financial cost for this additional resource by extrapolating their data to an entire hospital implementing numerous drug changes annually. Most administrators would require more evidence than this small sample size to justify changes in resource allocation.

It would have been interesting to have the researchers measure the time impact on nursing personnel, particularly in the outpatient rheumatology unit of switching brands of biologic medications from reference brand to biosimilar, as nursing education and support of patients is crucial for acceptance and adherence in that scenario.

6. PLOS authors have the option to publish the peer review history of their article (what does this mean?). If published, this will include your full peer review and any attached files.

Reviewer #1: No

Reviewer #2: No

---

## [Author Response · Author response to Decision Letter 0]

26 Nov 2020

Dear reviewers at PLOS ONE

Thank you for a thorough reading and constructive criticism of our manuscript entitled "Time spent by hospital personnel on drug changes: A time and motion study from an in-and outpatient hospital setting”. We are pleased with the opportunity to revise the original manuscript and we especially thank you for your relevant comments to the article. You have pointed out important aspects that needs clarifications, which we have tried our best to respond to. With these improvements, we hope to attract readers all over the health care sector, health authorities and the pharmaceutical industry. 

It is with great pleasure that we send this rebuttal letter that responds to your suggested revision of the original manuscript for resubmission to PLOS ONE. We hope the revised article meets your expectations satisfactorily. Again, we thank you for your professional opinion, constructive feedback and input. Enjoy the revised article.

On behalf of all co-authors,

Yours sincerely,

Joo Hanne Poulsen

Comments to the Author

1. Is the manuscript technically sound, and do the data support the conclusions?

We hope that the revised manuscript appears technically sound, and that the data support the conclusions. 

2. Has the statistical analysis been performed appropriately and rigorously? 

Answer to reviewer #1: Please see the detailed comments under section 5, item 6. 

3. Have the authors made all data underlying the findings in their manuscript fully available?

No comment

4. Is the manuscript presented in an intelligible fashion and written in standard English?

Thank you. 

5. Review Comments to the Author

Reviewer #1: 

1) The data has been collected by a single person in this study (JHP). This definitely leads to personal bias in all the data collection aspects. It would have been better if at least two members were involved in data collection stage.

Answer to reviewer #1, question 1: We agree that this is a limitation in the study, which will be discussed more clearly in the revised manuscript under “strength and limitations”. In continuation of #reviewer comment 4), these limitations have been clarified and discussed in the revised manuscript.

We hope that it meets your comment satisfactorily. 

2) The authors have mentioned that it would not be possible to measure the Hawthorne effect. But this is an experimental study which has been carried out using sophisticated techniques like Tobii eye tracker, Go pro camera etc. Unless the nurses and SHA are trained on these methods, the impact created by these equipment on the behaviour cannot be underestimated.

Answer to reviewer #1, question 2: We agree that the Hawthorne effect should not be underestimated. The use of an eye tracker in this study has affected the nurses and SHA, but the magnitude of the effect is unfamiliar, as nurses and SHA could be either faster or slower in the presence of an observer, eye tracking and go-pro camera. However, we have stated this limitation more clearly in the revised manuscript. 

“The extent of the Hawthorne effect is difficult to estimate in the study, but the observer is assumed to have affected the observed participants to some extent. However, the general impression is that this bias diminished over time, since the participants became more and more comfortable with being observed [32]. Additionally, the physical appearance of the go-pro camera and the eye-tracking glasses mounted to the participant’s head have indeed also affected the participants while being observed. However, just as the bias related to the observer’s presence, the general level of interference of the glasses is assumed to diminish over time as participants become more accustomed with the glasses. The Hawthorne effect in this matter is impossible to escape, unless the nurses and SHA became accustomed to working with the glasses”. 

We hope that it meets your comment satisfactorily. 

3) In the cardiology wing, data has been collected in March 2019 (baseline) and April-May 2019 (immediately after DCs are introduced). This is definitely not a good comparison period because the authors have taken a baseline period wherein the nurses and SHAs were very well versed with the existing drugs (March 2019), and the study period of April-May 2019 is a learning phase even for an experienced nurse/SHA. If the data was collected a few months after the drug change (DC), the actual impact of the time could have been assessed and concluded.

Answer to reviewer #1, question 3: Thank you for your observation. We would like to emphasize that the purpose of the study was to explore the time spent by hospital personnel in a drug change situation. Thus, we were specifically interested in this phase immediately after a DC, which is the phase where the personnel gets familiar with the new drug. Furthermore, we were interested in exploring, whether the DS phase would persist long term, but this was only explored in the outpatient clinic. However, we acknowledge that the meaning of a 

“DC situation” may be unclear. Thus, in the introduction section (line 74) we attempt to clarify what a DC situation embodies (changes to drugs new/unfamiliar to the dispensing personnel). 

“The focus of this study is the DC situation, meaning the actual changes to new drugs unfamiliar to the dispensing personnel”.

Further, in the discussion, an additional statement is applied to the fact that no follow-up from the cardiology ward was made.

“No follow-up at the cardiology ward took place and thus, claiming that the extra time spent on DCs are related to immediate expenses due to tender are not possible. However, the follow-up data from the rheumatology outpatient clinic indicates that the extra time observed in relation to DC diminished over time and returns to baseline. Thus, one could assume a similar trend at the cardiology ward as the nurses and SHA become more familiar with dispensing of the implemented drugs”.

We hope that our specifications meet your comment satisfactorily. 

4) In pages 7 & 8 of the manuscript, the authors mention how the software used for collecting data of Tobii glass tracker works. "Potential irrelevant times have been discarded". The authors need to explain more on how they decided irrelevant time. Moreover observing times from video and using it for data collection purpose can include 'bias'. Need to be addressed appropriately.

Answer to reviewer #1, question 4: We agree that there is a bias related to the use of observing time from video and using it as data. Further, having one researcher (JHP) performing the coding may have resulted in intra-rating bias. These limitations have been clarified and discussed in the revised manuscript. 

“At the cardiology ward, the same person (JHP) performed all the observations and coding of data. This may have resulted in personal and intra-rating bias, which could potentially affect the reliability of the study [40]. Obviously, having two or more researcher performing the observations and coding independently, discuss their finding or calculating a coder reliability, would enhance the overall reliability and reduce potential personal bias in the study. One way to minimize the potential intra-rating bias is to reduce the risk of “coder-drift”, which refers to applying the coding differently due to boredom or fatigue when coding larger data set at once [40]. However, in this study the coding was performed immediately after each observation and thus potentially reduce intra-rating bias [40]. Further, figure 1 displays the coding frame for the observations hereby increasing the transparency of the coding of included and discarded time. Conversely, having the same person performing the data collection results in less observation bias and reliability issues compared to the outpatient clinic, where pharmacy internship students were involved too. Despite the guidance of students, there might be differences in the perception of the time registrations in the consultation elements. However, these differences are considered to have minor influence on the registered time differences and overall findings”.

Additionally, based on your relevant comment, figure 1 has been added to increase the transparency of the data analysis. Figure 1 provides an example of the coding of relevant and irrelevant time intervals related to medicine dispensing at the cardiology ward. 

“The time division, also referred to as coding in this study, was performed immediately after each observation. An example of the coding of relevant and irrelevant time intervals related to medicine dispensing step 3-6 at the cardiology ward is provided in Fig 1”. See Fig 1. Coding of relevant and irrelevant time intervals – an example. An example of coding of relevant and irrelevant time intervals related to medicine dispensing at the cardiology ward.

We hope that our specifications and supplement of the table to the manuscript meet your comment satisfactorily.

5) The authors should explain why they have used different data collection techniques for inpatient (eye tracker, video etc.) and outpatient (mobile stop watch) setting.

Answer to reviewer #1, question 5: We agree with your comment. Hence, in the method section, the reasons for the different data collection techniques are highlighted further. 

“As the dispensing process at the cardiology ward involves seconds, the use of a refined and detailed data collection technique is required. Thus, the eye tracking was used in this setting. Additionally, the dispensing process at the rheumatology outpatient setting contain other activities and patients, thus, the use of an overall time measurement, such as a stopwatch was more useful in this setting. The use of the details provided by the eye tracking were not relevant in outpatient setting, just as the physical presence of the glasses would induce a considerable disturbance in the patient consultation. Thus, different data collecting technique were used for in- and outpatient settings”.

We hope that our specifications meet your comment satisfactorily.

6) I did a quick data analysis for cardiology ward (baseline) and cardiology ward (comparison).The data points for comparison period is very less. More over, the authors have taken an "average of averages" for the time comparison. This is definitely not a correct practise. For instance, even in base line phase average time taken by nurse 5 is 30.7 sec and that of nurse 7 is 14 seconds. So nurses cannot be treated as "homogeneous units" in data analysis. Need more explanation on this.

Answer to reviewer #1, question 6: Thank you for the comment. We acknowledge and agree that the nurses are not a homogeneous group, however; we are unsure what is meant by the “average of averages”. The supporting information contain raw data for baseline, comparison and follow-up (outpatient clinic) and from this raw data, averages are calculated. Thus, we do not calculate an average for one specific nurse and then calculate the averages for all nurses afterwards. Further, we are aware of the differences between the registered dispensing, thus, the standard deviation, as well as the minimum and maximum, are outlined in the manuscript. In these tables the variations between the different time points appear. Additionally, we do not focus on the time differences between the individual nurses at the cardiology ward. It is not of relevance to the study to adjust for the individual nurse before and after the DCs, as this would entail the dispensing of the exact same drug in both periods, which is not measured and thus not focused on in this study. 

We are happy to present data in another way, however; a clarification of the comment in further details would be needed for us to comply to the comment satisfactorily. 

Reviewer #2: Comments have been pasted from a Word document (attached) as info was lost due to the way the system saves info. Data provided support the conclusions, however the data set is quite limited.

Answer to reviewer #2: We fully agree with the comment and we have stated that more clearly in the manuscript in terms of affecting the reliability of results. 

“Even though the actual implementation of DCs were effective from April 1st, relatively few DCs were observed in the cardiology ward in the comparison period (April and May), which potentially affects the reliability of the results”.

We hope that our specifications meet your comment satisfactorily.

The authors have taken a non-biased approach to measuring time impact, using eye-tracking technology rather than just direct observation – this is commendable. The appropriately cite the difficulty with other reports of time impact of drug changes using survey methods, which are affected by recall bias. Unlike most other publications that focus on pharmacy activities, this report focuses on nursing activity and the impact of drug changes. This situation may be more relevant to hospitals that put more drug control in the hands of pharmacy personnel and that use more automated systems, such as Automated Dispensing Units, barcode technology and even unit dose medications. Most North American institutions have medication orders written by generic name and drug selection is not required by nursing personnel. Thus drug changes (in brand provided) do not impact nursing time. The changes reported by the authors were not significant in all categories, with only semi-drug changes causing a significant increase in time spent by nursing – these changes were relatively short-lived, with time spent returning almost to baseline. IN the outpatient setting of rheumatology clinic, time was impacted more significantly, but for a brief interval only for drug changes and also for drug shortage situations, which may occur at any time of year, regardless of contract changes.

The authors suggest (lines 282-286) that system changes such as prescribing using generic names could prevent confusion during drug changes. Inventory management was briefly alluded to as the medication room continued to have stock of the previous brand, leading to incongruence between prescribed drug name and available drug name.

The authors suggest that administrators should be made aware of the need for additional time (resource) to adapt to drug changes, but they have not provided a financial cost for this additional resource by extrapolating their data to an entire hospital implementing numerous drug changes annually. Most administrators would require more evidence than this small sample size to justify changes in resource allocation.

Answer to reviewer #2: We fully agree with the comment. Thus, we have calculated and thus transitioned the reportable time spent on dispensing and extradition of medicine from in- and outpatient settings into economical expenses in USD. Details and additional sections are provided in order to calculate the economic impact. Thus, they are found in the method section for both sub-studies, a new economic impact section (in method and results), just as the discussion and conclusion of the economic impact is changed accordingly to the results. 

We hope that this supplement meets your comment satisfactorily.

It would have been interesting to have the researchers measure the time impact on nursing personnel, particularly in the outpatient rheumatology unit of switching brands of biologic medications from reference brand to biosimilar, as nursing education and support of patients is crucial for acceptance and adherence in that scenario.

Answer to reviewer #2: We agree with this perspective. It would have been highly interesting to explore DCs of biosimilar medicines from the perspectives of nursing and patient education, but the focus of this study was on general DCs due to tendering. A comment in the discussion section containing DC from originator to biosimilar are included. However, another study would be required in order to explore this perspective in details. 

“Further, this study focusses on a generic DC due to tender, but one could image that dealing with DC from originator to biosimilar would require extra resources for nursing education and patient information, due to another complexity of the DC. However, another study would be required in order to explore this perspective in details”.

We hope that this supplement meets your comment satisfactorily.

---

## [Decision Letter · Decision Letter 1]

21 Dec 2020

PONE-D-20-21353R1

Time spent by hospital personnel on drug changes: A time and motion study from an in-and outpatient hospital setting

PLOS ONE

Dear Dr. Poulsen,

Thank you for submitting your manuscript to PLOS ONE. After careful consideration, we feel that it has merit but does not fully meet PLOS ONE’s publication criteria as it currently stands. Therefore, we invite you to submit a revised version of the manuscript that addresses the points raised during the review process.

Please ensure that you address comments 1, 3 and 4 of reviewer 1. Please respond to comment 2 of reviewer 1 as well as possible given the data that you have.

We look forward to receiving your revised manuscript.

Kind regards,

Susan Horton

Academic Editor

PLOS ONE

Additional Editor Comments (if provided):

Reviewer 2 is satisfied with the changes made; I hope you can respond to Reviewer 1's remaining questions.

Reviewers' comments:

Reviewer's Responses to Questions

**Comments to the Author**

1. If the authors have adequately addressed your comments raised in a previous round of review and you feel that this manuscript is now acceptable for publication, you may indicate that here to bypass the “Comments to the Author” section, enter your conflict of interest statement in the “Confidential to Editor” section, and submit your "Accept" recommendation.

Reviewer #1: (No Response)

Reviewer #2: All comments have been addressed

2. Is the manuscript technically sound, and do the data support the conclusions?

Reviewer #1: Partly

Reviewer #2: Yes

3. Has the statistical analysis been performed appropriately and rigorously? 

Reviewer #1: I Don't Know

Reviewer #2: Yes

4. Have the authors made all data underlying the findings in their manuscript fully available?

Reviewer #1: Yes

Reviewer #2: Yes

5. Is the manuscript presented in an intelligible fashion and written in standard English?

Reviewer #1: Yes

Reviewer #2: Yes

6. Review Comments to the Author

Reviewer #1: I have gone through the comments that you have provided. There are a few unanswered questions.

1) Can you please clarify the objective of the study? If the focus is only on seeing the time change during DC, it is very important to explain the duration of this effect? For example, say if a new drug change has happened on April 1st, till what time do you observe this increased drug dispensing time? Is it till May 1st or June 1st etc..

2) You mention that ". The extent of the Hawthorne effect is difficult to estimate in the study, but the observer is assumed to have affected the observed participants to some extent. However, , but the general impression is that this bias diminished over time, since the participants became more and more comfortable with being observed"?

What is the meaning diminished over time? This is again connected to previous question. If this claim has to be made, then you need to have the following data. Before drug change, identify the base line time for dispensing with and without observer. Quantify the difference between the two. Once you notice that there is no statistical difference in the base line time of dispensing for baseline (with and without observer), you need to look for effect in situations of drug change. I comment this based on your results in Table 3. For baseline, you have 20.5 +/- 10.5 as the time. However after DC it is 28.4 +/- 16.3. Higher standard deviation can be due to presence of observer or due to the change in drugs. This is definitely not a good thing. Again, you do not mention how many days did this increased time (28.4 sec) lasted after DC.

3) Figure 1 is unclear. Please provide a better quality picture and an example of timelines for a particular scenario.

4) To add further, since I noticed that the standard deviation is higher in DC, I wonder if it would be due to the effect of uncertainty in the measurements in itself. Can you please provide how much is the uncertainty of the measurement system?

Reviewer #2: The authors have addressed my comments very well. The paper reads well and the authors have made adjustments based on other reviewer comments also. The strengths and limitations section as well as economic impact have been enhanced.

7. PLOS authors have the option to publish the peer review history of their article (what does this mean?). If published, this will include your full peer review and any attached files.

Reviewer #1: No

Reviewer #2: **Yes: **Lauren F. Charbonneau, RPh, BScPharm, Manager, Pharmacy Sunnybrook Odette Cancer Centre

---

## [Author Response · Author response to Decision Letter 1]

3 Feb 2021

Dear reviewer #1 at PLOS ONE

Once again, thank you for a thorough reading and constructive criticism of our manuscript entitled "Time spent by hospital personnel on drug changes: A time and motion study from an in-and outpatient hospital setting”. We thank you for your comments to the article, and we have tried our best to respond to the aspects that needs further clarifications. 

It is with great pleasure that we send this rebuttal letter that responds to your suggested revision of the original manuscript for resubmission to PLOS ONE. We hope the revised article meets your expectations satisfactorily. 

On behalf of all co-authors,

Yours sincerely,

Joo Hanne Poulsen 

Reviewers' comments:

Reviewer's Responses to Questions

Comments to the Author

1. If the authors have adequately addressed your comments raised in a previous round of review and you feel that this manuscript is now acceptable for publication, you may indicate that here to bypass the “Comments to the Author” section, enter your conflict of interest statement in the “Confidential to Editor” section, and submit your "Accept" recommendation.

Reviewer #1: (No Response)

Reviewer #2: All comments have been addressed

We hope that the revised manuscript is now acceptable for publication. 

2. Is the manuscript technically sound, and do the data support the conclusions?

Reviewer #1: Partly

Reviewer #2: Yes

We hope that the revised manuscript appears technically sound, and that the data support the conclusions

3. Has the statistical analysis been performed appropriately and rigorously? 

Reviewer #1: I Don't Know

Reviewer #2: Yes

We believe that the statistical analysis that have been performed are appropriate and relevant and we hope that the revised manuscript is found acceptable for publication. ________________________________________

4. Have the authors made all data underlying the findings in their manuscript fully available?

Reviewer #1: Yes

Reviewer #2: Yes

5. Is the manuscript presented in an intelligible fashion and written in standard English?

Reviewer #1: Yes

Reviewer #2: Yes

6. Review Comments to the Author

Reviewer #1: I have gone through the comments that you have provided. There are a few unanswered questions.

1) Can you please clarify the objective of the study? If the focus is only on seeing the time change during DC, it is very important to explain the duration of this effect? For example, say if a new drug change has happened on April 1st, till what time do you observe this increased drug dispensing time? Is it till May 1st or June 1st etc.

Thank you for your comment. We would like to emphasize that the aim of the study is to explore time spend by hospital personnel in a drug change situation when dispensing medicine to in- and outpatients in a hospital setting in the Capital Region of Denmark. Thus, the focus of this study is on the specific DC situation, meaning the time where the drugs are actually changing to new drugs unfamiliar to the dispensing personnel. We acknowledge that it would be an interesting aspect to know the duration of the effect, but our study was not designed to answer this question. 

In this study, we have thus not explored the length or duration of the period of ‘semi-drug change’ and DC. Therefore, we are unable to answer how long the duration of the effect on dispensing time that we have observed is. In the study, we assume a 4 week period of ‘semi-drug change’ and DC when calculating the economic impact, but the actual duration of the periods are unknown with our dataset. Thus, based on your comment, we will emphasize this further in the discussion of the economic impact (line 370-380). 

At the cardiology ward, a total of 27 nurses and SHA’s cover the dispensing process in both day and evening shifts. Estimating that ‘semi-drug change’ and DC affect dispensing time in a period of 4 weeks leads to a calculated extra time of 2 hours and 22 minutes or 196 USD spent on the assumed ‘semi-drug change’ and DC at this particular cardiology ward. This extra time and expenses does not immediately seem to have a clinical and economic impact, especially not on a daily basis with extra time corresponding to 303 seconds and 7 USD. However, the size of the cardiology ward is assumed to cover a standard medical ward in the Capital Region of Denmark. Thus, multiplication of the expenses with the total number of medical wards will show a considerably amount of additional expenses spent locally, regionally or even nationally. It should be emphasized that the estimated 4 weeks period with ‘semi-drug change’ and DC are based on assumptions, since the actual duration of the periods are unknown in this study.

We hope that our specifications meet your comment satisfactorily. 

2) You mention that ". The extent of the Hawthorne effect is difficult to estimate in the study, but the observer is assumed to have affected the observed participants to some extent. However, , but the general impression is that this bias diminished over time, since the participants became more and more comfortable with being observed"?

What is the meaning diminished over time? This is again connected to previous question. If this claim has to be made, then you need to have the following data. Before drug change, identify the base line time for dispensing with and without observer. Quantify the difference between the two. Once you notice that there is no statistical difference in the base line time of dispensing for baseline (with and without observer), you need to look for effect in situations of drug change. I comment this based on your results in Table 3. For baseline, you have 20.5 +/- 10.5 as the time. However after DC it is 28.4 +/- 16.3. Higher standard deviation can be due to presence of observer or due to the change in drugs. This is definitely not a good thing. Again, you do not mention how many days did this increased time (28.4 sec) lasted after DC.

Thank you for your comment. We have chosen to delete and re-write the sentence to meet your comment (line 418-428). We do not measure the Hawthorne effect, and therefore we do not know whether or not it diminishes over time, thus we simply wishes to acknowledge that it is theoretically present. However, as the same participants and observer is present in baseline and comparison, we assume that the Hawthorne effect is equally widespread in both data collection periods. In addition to this assumption, we have not changed any procedures or done anything differently in the data collection periods that can either minimize, maximize or affect the Hawthorne effect from one period to another. 

The extent of the Hawthorne effect is difficult to estimate in the study, but the observer is assumed to have affected the observed participants to a minor extent. Additionally, the physical appearance of the go-pro camera and the eye-tracking glasses mounted to the participant’s head have indeed also affected the participants while being observed. The Hawthorne effect in this matter is impossible to avoid, unless the nurses and SHA became accustomed to working with the glasses. Further, as the same participants and observer is present in baseline and comparison, the Hawthorne effect is assumed equally widespread in both data collection periods, although the effect has not been measured in this study.

In terms of the standard deviations (SD) in baseline versus comparison, we acknowledge that there is a difference in SD’s. Based on the already assumed Hawthorne effect in both data collection periods, we do not assume that the Hawthorne effect has had a significant impact on SD. With regard to DCs, we agree that the effect on both mean and SD can be due to the change in drugs or ‘semi-drug change’. Another relevant factor that may impact the SD is the sample size. In baseline, the sample size is N = 161, whereas it is N = 35 (DCs and ‘semi-drug changes’) in comparison, and thus the calculation of SD in comparison is based on a relatively small sample size, which implies a larger SD. We have not included this in the discussion as it is a statistical technicality and the study was not designed to collect generalizable data and thus no power calculation were performed before data collection. 

With regard to the duration of the effect on dispensing time we hope that our provided specifications meet the comment by Reviewer #1 satisfactorily. 

3) Figure 1 is unclear. Please provide a better quality picture and an example of timelines for a particular scenario.

Thank you for your observation. Smaller corrections to figure 1 have been made in order to demonstrate more clearly how the coding of relevant and irrelevant time intervals related to dispensing at the cardiology ward have been divided and the time included or discarded in the study. An example of a time line for a particular scenario covering the dispensing process step 3-6 for Hexalactone has been provided (line 157-162). 

An example of the coding of relevant and irrelevant time intervals related to medicine dispensing in the cardiology ward for a particular scenario covering the dispensing process step 3-6 for Hexalactone has been provided in Fig 1.

Fig 1. Coding of relevant and irrelevant time intervals – an example. An example of coding of relevant and irrelevant time intervals related to medicine dispensing at the cardiology ward for a particular scenario covering the dispensing process step 3-6 for Hexalactone. 

We hope that the quality of figure 1 is more clear now and that the amendments meet the reviewers comment satisfactorily. 

4) To add further, since I noticed that the standard deviation is higher in DC, I wonder if it would be due to the effect of uncertainty in the measurements in itself. Can you please provide how much is the uncertainty of the measurement system?

Thank you for the comment. We are not entirely sure what is meant with the “uncertainty of the measurement system”. Throughout the data collection, the same measurement methods in the in- and outpatient setting respectively have been used. Thus, we assume that the measurement methods hold the same uncertainty in both baseline and comparison.

Reviewer #2: The authors have addressed my comments very well. The paper reads well and the authors have made adjustments based on other reviewer comments also. The strengths and limitations section as well as economic impact have been enhanced.

Thank you very much for review and constructive feedback.

---

## [Editor Report · Decision Letter 2]

9 Feb 2021

Time spent by hospital personnel on drug changes: A time and motion study from an in-and outpatient hospital setting

PONE-D-20-21353R2

Dear Dr. Poulsen,

We’re pleased to inform you that your manuscript has been judged scientifically suitable for publication and will be formally accepted for publication once it meets all outstanding technical requirements.

Kind regards,

Susan Horton

Academic Editor

PLOS ONE

Additional Editor Comments (optional):

Thank you for addressing the remaining reviewer comments.
---

## [Editor Report · Acceptance letter]

12 Feb 2021

PONE-D-20-21353R2 

Time spent by hospital personnel on drug changes: A time and motion study from an in-and outpatient hospital setting 

Dear Dr. Poulsen:

I'm pleased to inform you that your manuscript has been deemed suitable for publication in PLOS ONE. Congratulations! Your manuscript is now with our production department. 

Kind regards, 

on behalf of

Dr. Susan Horton 

Academic Editor

PLOS ONE